# Spatio-temporal dynamics of hand, foot and mouth disease in Malaysia, 2009–2019

Victoria M. Cox[1], Rohani Jahis[2], Rehan Shuhada Abu Bakar[3], Noriah Mohd Yusof[3], Kiroshika Pillay Veel Pilay[3], Yu Kie Chem[3], S. Selvanesan[3], Norita Shamsudin[2], Wes Hinsley[1], Isobel M. Blake[1], Nor Zahrin Hasran[3], Norhayati Rusli[2], I-Ching Sam[4], Yoke Fun Chan[4]*, Margarita Pons-Salort●[1]*

**1** MRC Centre for Global Infectious Disease Analysis, School of Public Health, Imperial College, London, United Kingdom, **2** Disease Control Division, Ministry of Health, Putrajaya, Malaysia, **3** National Public Health Laboratory, Ministry of Health, Sungai Buloh, Malaysia, **4** Department of Medical Microbiology, Faculty of Medicine, Universiti Malaya, Kuala Lumpur, Malaysia

* chanyf@um.edu.my (YFC); m.pons-salort@imperial.ac.uk (MPS)

## Abstract

Hand, foot and mouth disease (HFMD) is endemic in Asia-Pacific. There is geographic variability in the timing of HFMD outbreaks throughout the year across the region, and seasonality becomes less clear towards tropical regions. We used syndromic HFMD case surveillance data from 2009 to 2019 (N = 354,769 cases) to characterise the spatio-temporal dynamics of HFMD in Malaysia, which has a tropical climate, and identify factors associated with transmission. Despite the absence of regular annual seasonal patterns of incidence, HFMD epidemics were highly synchronised across districts within Peninsular and East Malaysia, but less so between the two regions. Median estimates of the state-level daily effective reproduction number (Rt) ranged from 0.47 to 1.54. Meteorological factors were found to have a small effect on HFMD transmission compared to the depletion of susceptibles (as a proxy for population immunity) and school closures, likely due to the low seasonal weather variability across the year. Studies using wider spatial scales covering a diversity of climate regions are needed to identify meteorological factors determining the timing of HFMD epidemics across Asia-Pacific countries.

## Author summary

Hand, foot and mouth disease (HFMD) is a common viral infection primarily affecting young children. It is caused by a few enteroviruses, which are mostly transmitted through the faecal-oral route. HFMD outbreaks have been reported extensively across the Asia-Pacific region over the last twenty years, imposing an important burden on the healthcare systems. Here, we analysed HFMD case surveillance data over an 11-year period before the COVID-19 pandemic

**Data availability statement:** Hand, foot and mouth disease (HFMD) surveillance data is curated and stored by the National HFMD Surveillance System of Malaysia, under the Ministry of Health and available on the government website data.org.my at https://archive.data.gov.my/data/ms_MY/dataset/bilangan-kes-penyakit-hfmd-mingguan-mengi-kut-negeri.

**Funding:** The authors acknowledge funding from the MRC Centre for Global Infectious Disease Analysis (reference MR/X020258/1), funded by the UK Medical Research Council (MRC). This UK-funded award is carried out in the frame of the Global Health EDCTP3 Joint Undertaking. M.P.-S. is a Sir Henry Dale Fellow jointly funded by the Wellcome Trust and the Royal Society (grant number 216427/Z/19/Z). V.M.C. acknowledges funding from the Wellcome Trust (grant 222375/Z/21/Z). The funders had no role in study design, data collection and analysis, decision to publish, or preparation of the manuscript.

**Competing interests:** The authors have declared that no competing interests exist.

(2009–2019) in Malaysia, to explore the dynamics and drivers of HFMD epidemics. We found that HFMD is endemic and does not exhibit a regular seasonal pattern in Malaysia, contrary to other countries, where outbreaks tend to occur in summer. HFMD epidemics were highly synchronised across districts within Peninsular and East Malaysia, but less so between these two regions. Meteorological factors had a very small effect on transmission, potentially because of the low weather variability across the year in Malaysia, given its tropical climate. The depletion of susceptible individuals in the population and school closures were the factors most strongly associated with HFMD transmission.

## Introduction

Hand, foot and mouth disease (HFMD) is a common childhood illness endemic across the Asia-Pacific region with thousands to millions of cases reported each year in some countries [1–3]. The main symptoms of HFMD are fever and a skin rash with blisters in the mouth and on the hands and feet [4]. The disease is most prevalent in children aged under five years [1] and is caused by several enteroviruses that are mostly transmitted through the faecal-oral route [5]. Enterovirus A71 (EV-A71), coxsackievirus A16 (CVA16), and coxsackievirus A6 (CVA6) are the serotypes most commonly reported in HFMD cases [6,7]. Most children have mild symptoms, but in a minority of cases (usually attributed to EV-A71), HFMD progresses to neurological and cardiopulmonary complications that can be fatal [5].

HFMD outbreaks are highly seasonal and tend to peak in summer in temperate regions, but there is high geographic variability in the timing of HFMD outbreaks throughout the year [1,8]. In northern regions of China there are regular annual peaks in June compared to in southern regions where there are biannual peaks occurring in May and September/October [1]. In the USA, enterovirus seasonality also shows a clear geographical pattern, with the distribution of cases having a less pronounced peak in the south and a more pronounced one towards the north [8]. Across Asia, in temperate countries such as Japan, HFMD outbreaks and other enterovirus-related diseases tend to peak in July [9], whereas seasonality is less clear in tropical and subtropical regions – for example, Hong Kong and Singapore can experience multiple HFMD outbreaks in a year and the timing of outbreaks is not consistent across years [10,11]. Malaysia experiences frequent large-scale outbreaks of HFMD, however the timing and size of the outbreaks does not follow a regular seasonal pattern [2].

Several studies have shown that the main driver of enterovirus epidemic dynamics is the depletion of susceptible individuals [2,9,10,12], with serotype-specific population immunity and births driving the observed long-term cycles (over time periods of several years) of individual enterovirus serotypes [9,12]. However, there is currently no consensus on the main meteorological, demographic, or other factors that could explain differences in the seasonal pattern (i.e., timing of epidemics within the year) and absence of it, of HFMD epidemics across regions. Dai *et al*. found that the timing

of the start of school semesters was the main factor driving multiple annual peaks of HFMD in Wenzhou, China, and that meteorological factors during the spring school semester were also important [13]. This contrasts to Yang *et al.*'s findings for Hong Kong, that both school holiday timings and meteorological variables had only a small impact on HFMD transmission, compared to depletion of susceptibles [10].

Despite a good national surveillance system in Malaysia, the number of HFMD cases remains high and HFMD is one of the most common infectious diseases. For example, in 2022, it was second in number of reported cases behind SARS-CoV-2 [14]. Although vaccines protecting against EV-A71 exist and are licensed in China [15], they are not available in Malaysia, where outbreak management strategies rely on contact tracing, health education campaigns and school closures [4].

In this study, we explore the spatio-temporal distribution of HFMD cases in Malaysia using data collected through the national surveillance system between January 2009 and December 2019 across Peninsular Malaysia and East Malaysia. We estimate HFMD transmission through inference of its effective reproduction number and study the association between transmission and a series of meteorological factors and school holidays.

## Results

### Demographic characteristics of HFMD cases

A total of 354,769 HFMD cases were reported to the national surveillance system of Malaysia between January 1, 2009, and December 31, 2019, of which 237,038 (67%) were from Peninsular Malaysia and 117,731 (33%) from East Malaysia. Among those, 56% were males and 44% were females. 85% of the reported cases with age information (96.6%) were <= 5-years old. The number of cases was highest among those aged 12–23 months and declined afterwards (Fig 1A); however, the number of cases increased quickly with age during the first year of life (Fig 1B). The mean age of cases was 3.9 years, with a median age of 2.6 years and an interquartile range of 3.1 years.

### HFMD incidence, synchronisation and lack of seasonality

The daily number of recorded HFMD cases between January 1, 2009, and December 31, 2019, in both Peninsular and East Malaysia is shown in Fig 2A. In Peninsular Malaysia, the number of cases during the first three years of the study period was relatively low compared to the following years. In both regions, HFMD epidemics were of variable size and showed substantial variation in the timing of the peaks across the year, with epidemics generally more peaked in East

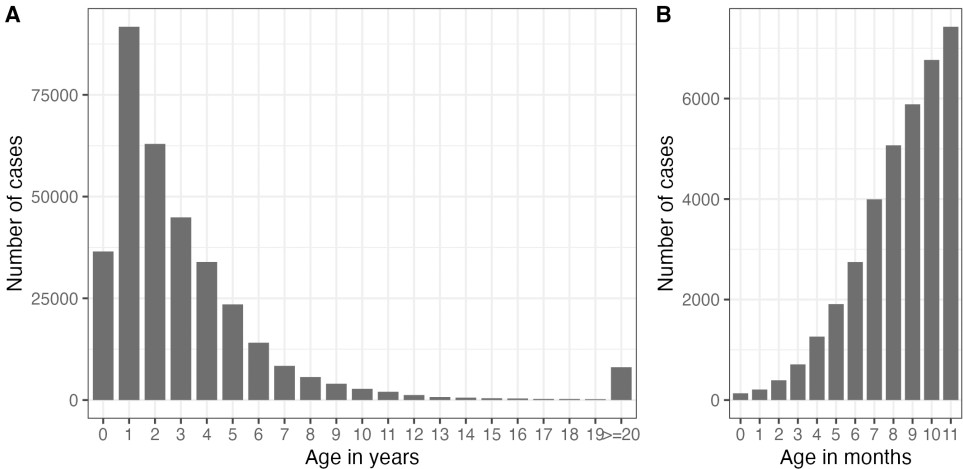

**Fig 1. Age distribution of HFMD cases in Malaysia reported between January 1, 2009, and December 31, 2019.** (A) Age distribution of cases in years. (B) Age distribution of cases in months among those less than one year old.

**A**

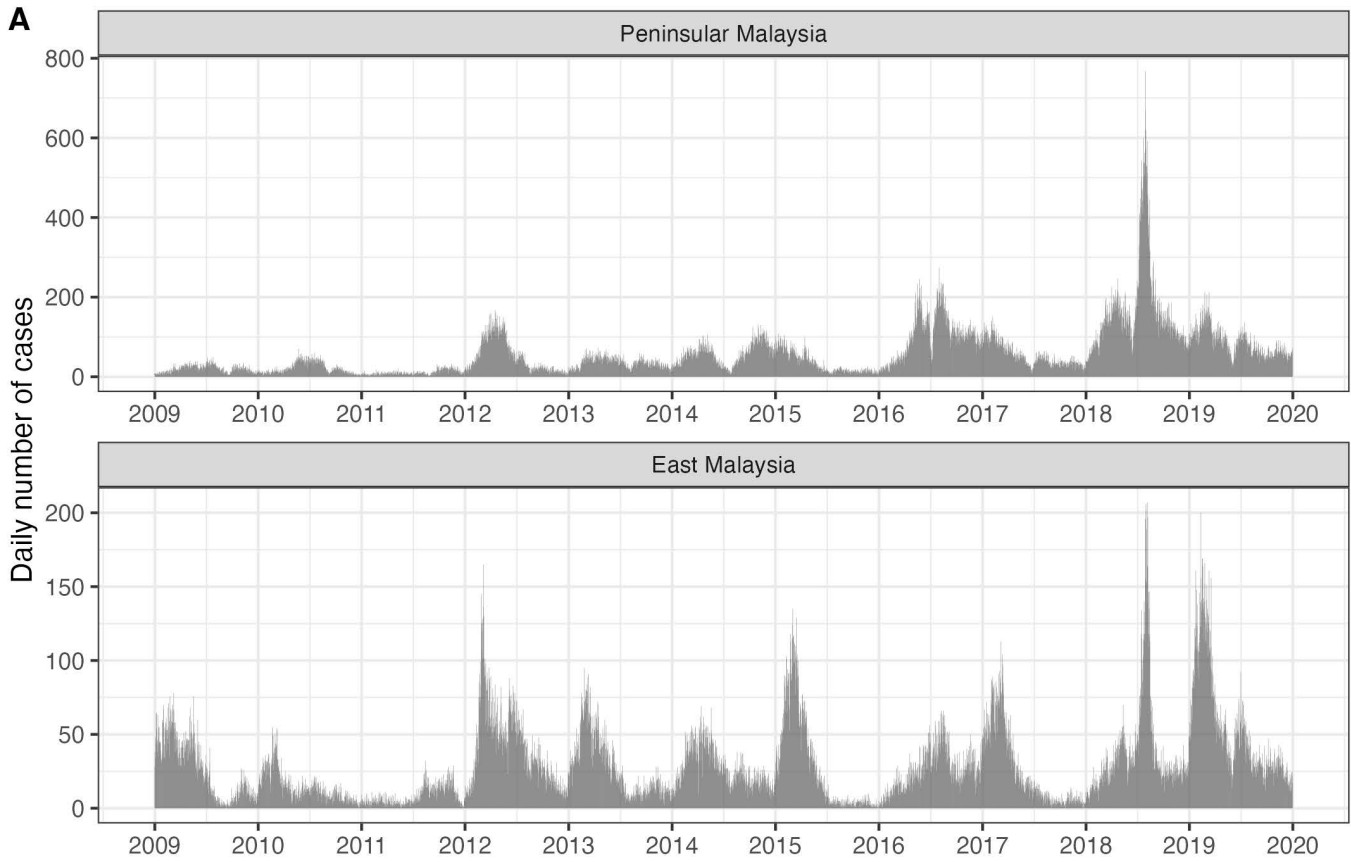

**B**

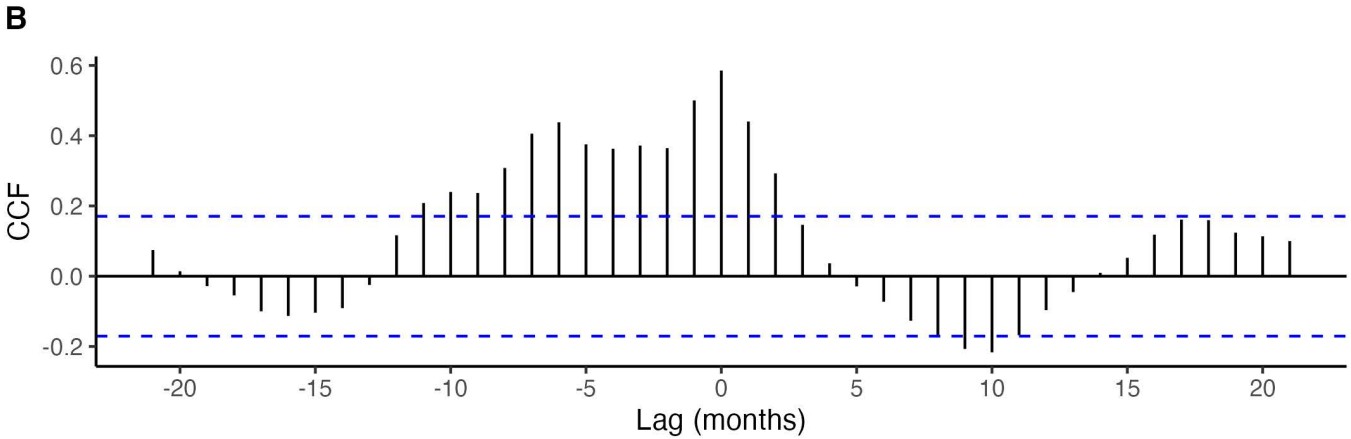

**Fig 2. Number and correlation of reported HFMD cases in the timeseries.** (A) Daily number of HFMD cases reported in Peninsular and East Malaysia between January 1, 2009, and December 31, 2019. (B) Cross-correlation (CCF = cross correlation function) between the timeseries of monthly number of cases in Peninsular and East Malaysia. The blue dashed lines indicate a significance level of 5%.

Malaysia. The largest HFMD outbreak occurred in 2018 in both Peninsular and East Malaysia, with a maximum number of cases reported in a single day of 768 and 207 cases respectively in Peninsular and East Malaysia. Interestingly, there was not a single day when no cases were reported in both regions.

Although a clear synchronisation of the epidemics across Peninsular and East Malaysia is difficult to assess visually, a cross-correlation of the two timeseries of the number of monthly cases indicates the strongest positive correlation at zero months, suggesting at least a moderate degree of synchronisation (Fig 2B). A second peak in the correlation function appears at a lag of six months, perhaps reflecting the existence of a semi-annual periodicity in the timeseries of cases.

A wavelet analysis revealed a 2-year period from around 2012 onwards in both regions, Peninsular and East Malaysia (Fig 3), indicative of some similarities in the long-term epidemic dynamics across the two regions. However, the same analysis also found an additional annual period in East Malaysia that was not obvious across the study period in Peninsular Malaysia.

The annual incidence of HFMD in East Malaysia (range 546–2845 cases per million people) was higher than in Peninsular Malaysia (range 347–2401 cases per million people) each year, except in 2016 and 2018 (Fig 4A). In Peninsular Malaysia, the annual HFMD incidence generally increased over time (Fig 4A), with the highest incidence observed in 2018.

Incidence over time by district shows a strong synchronisation of epidemics within both Peninsular and East Malaysia (Fig 4B), with many districts not reporting cases during the first three years. Of the 143 districts, from January 2012 (the year from which we observe larger outbreaks) through to December 2019, 24 districts reported HFMD cases every month (Fig 4C), of which five were in East Malaysia (Kota Kinabalu, capital of Sabah state; Kuching, capital of Sarawak state; Miri; Penampang, bordering Kota Kinabalu; and Sibu) and 19 were in Peninsular Malaysia (Fig 4C). The latter includes the Federal Territory of Kuala Lumpur (the national capital of Malaysia) and Johor Bahru (which borders Singapore). In contrast, there were 67 districts which reported zero cases over a period of at least three consecutive months between

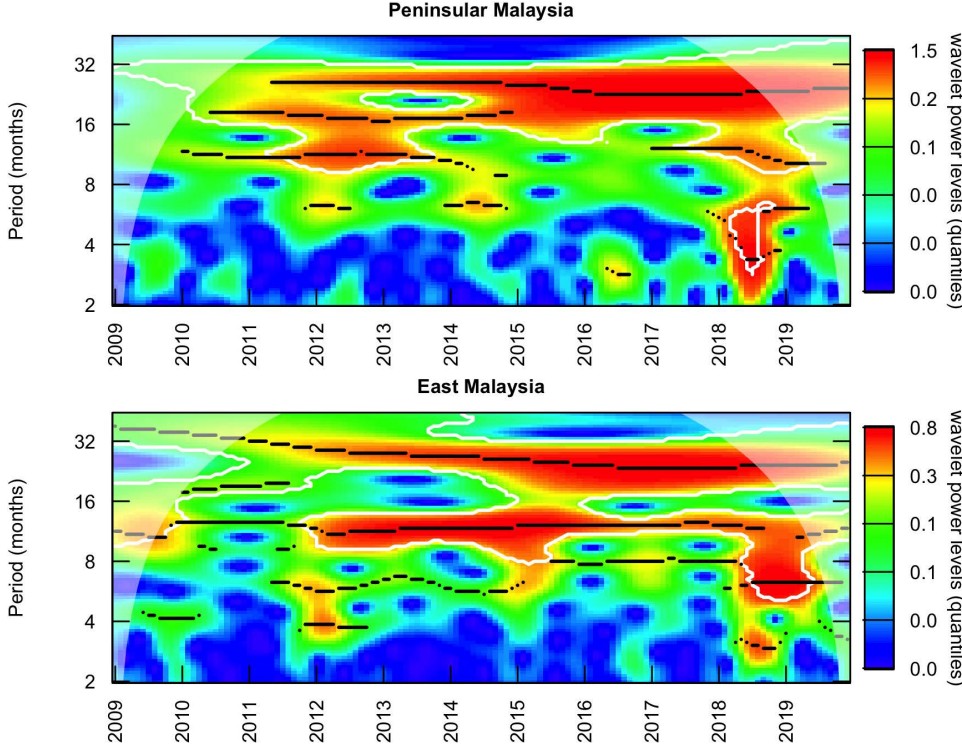

**Fig 3. Wavelet analysis.** Average wavelet power of the timeseries of number of HFMD reported cases by month (transformed to log (cases + 1)) in Peninsular Malaysia (top) and East Malaysia (bottom).

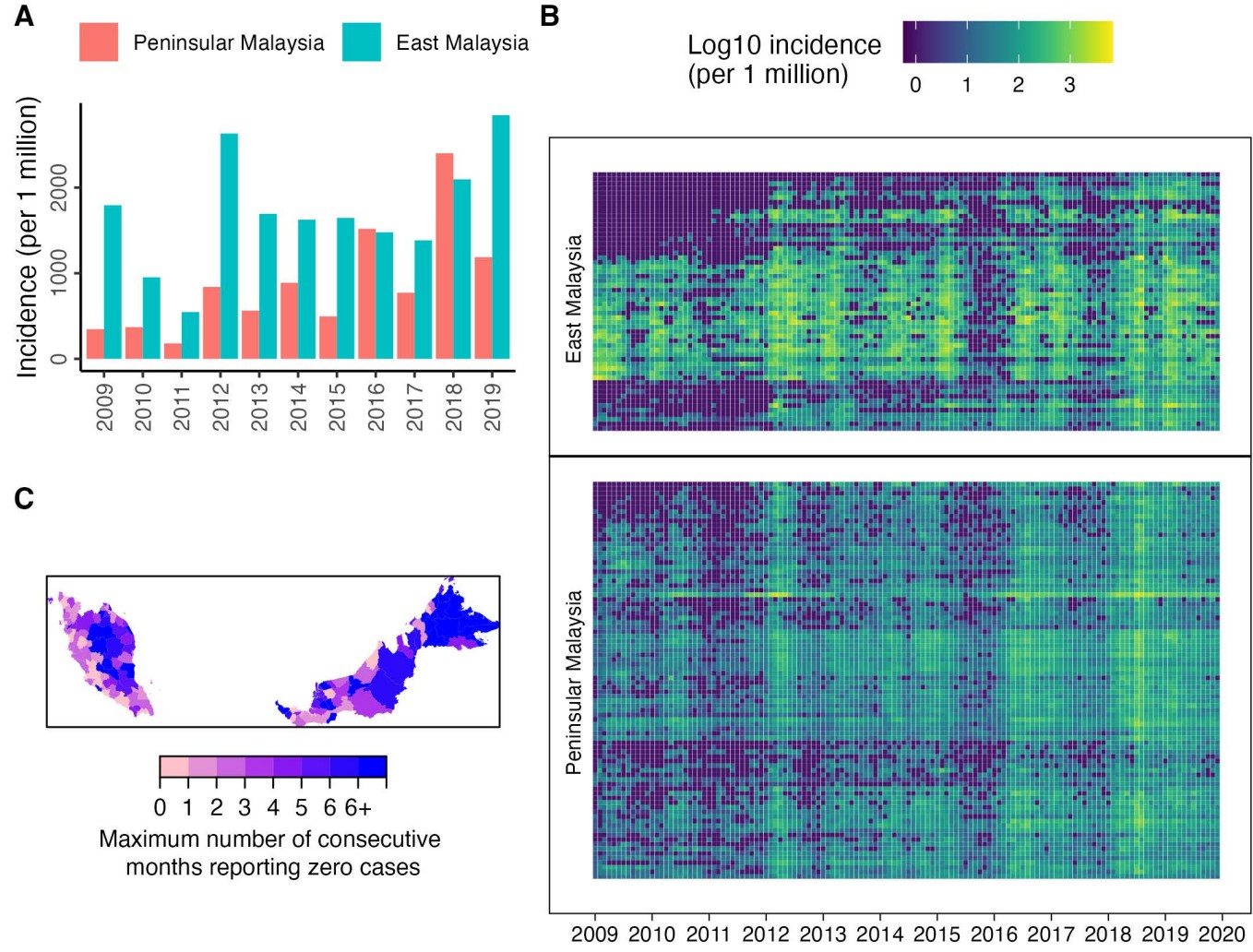

**Fig 4. Incidence rates.** (A) Annual incidence rates per million people in Peninsular and East Malaysia. (B) Monthly incidence rates per million people in each district, with districts split by Peninsular and East Malaysia and then ordered by latitude from top to bottom. (C) Maximum number of consecutive months reporting zero cases between January 2012 and December 2019 per district. The base layer of the map was sourced from GADM (https://gadm.org/download_country.html and https://gadm.org/license.html).

January 2012 and December 2019, and 30 of those had a minimum period of six months where they reported zero cases (Fig 4C). The maximum number of consecutive months a district reported zero cases for was negatively correlated with the population size of the district (S4 Fig).

The mean timing of all cases reported across the study period was similar in the three states in East Malaysia (estimated in the range between weeks 12 and 15) and different from the mean timing of cases in Peninsular Malaysia (estimated in the range between weeks 19 and 29) (Fig 5A). Within Peninsular Malaysia, similar estimates of the mean timing of cases were observed for neighbouring states, with a general latitudinal gradient (Fig 5). The distribution of all reported HFMD cases over the study period within the year displayed strong similarities for the states within each region (Peninsular Malaysia and East Malaysia), but some differences between the two regions (Figs 6A and S5): all states exhibited a peak within weeks 25 and 35, but only states in East Malaysia exhibited an earlier peak between weeks five and 15.

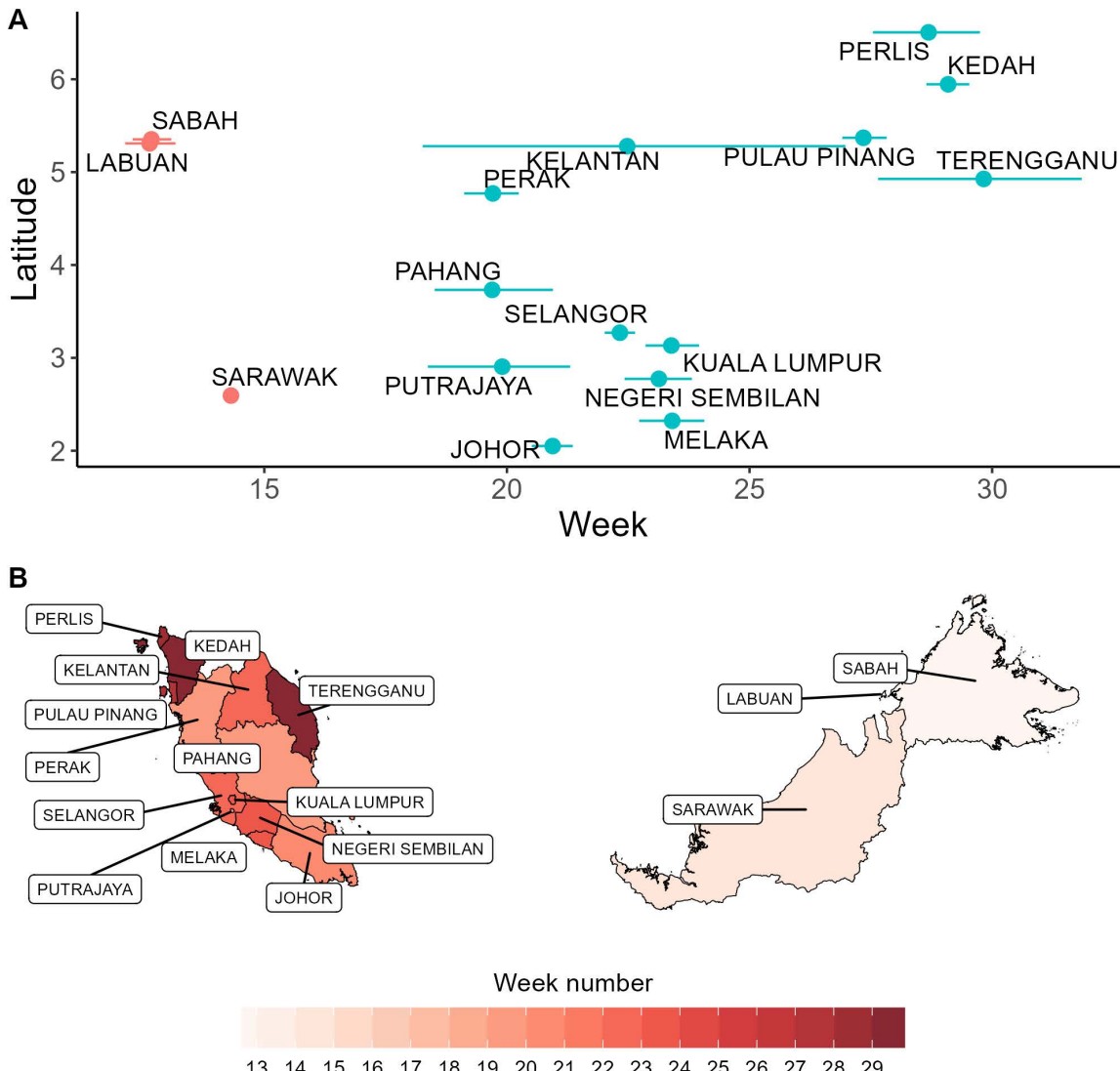

**Fig 5. Mean timing of cases.** (A) Mean timing of cases (in week number within the year), in each state as a function of latitude, taken as the centroid of its borders. (B) Mean timing of cases in each state shown in a map of Malaysia. The base layer of the map was sourced from GADM (https://gadm.org/download_country.html and https://gadm.org/license.html).

When looking at the weekly distribution of cases for each year of data independently, we found substantial differences across years within a state (see examples for the Federal Territory of Kuala Lumpur and Sarawak in Fig 6B, and all states in S6 Fig) which suggests that the seasonal pattern of HFMD epidemics is not regular every year.

### HFMD effective reproduction number

We estimated the daily effective reproduction number in each state (S7–S10 Figs). Median estimates of Rt fluctuated around one in the range of 0.47 to 1.54, with a short interquartile range of 1.01 to 1.14 (S11 Fig), suggesting HFMD is endemic and transmission occurs almost all year-round in most states. The lower bound of the 95% CrI was above one only on a few occasions, most of them at the beginning of the large 2018 outbreak.

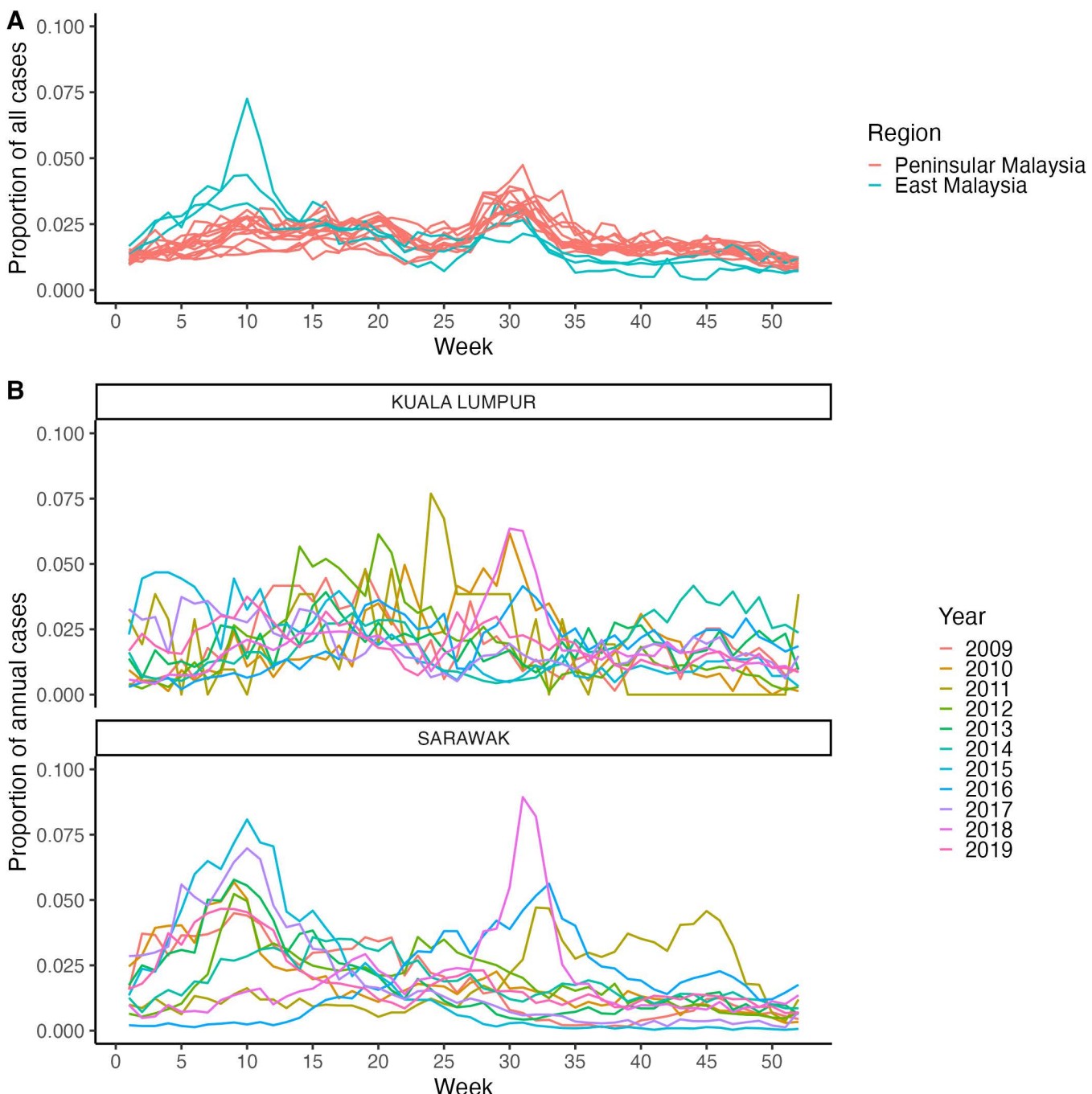

**Fig 6. Distribution of cases within the year.** (A) Weekly distribution of all cases reported between January 1, 2009, and December 31, 2019, within the year in each state. Each line corresponds to a state, and the lines are coloured according to their region, Peninsular and East Malaysia. (B) Weekly distribution of annual cases within the year for all years between 2009 and 2019 in the Federal Territory of Kuala Lumpur and Sarawak.

**Table 1. Estimated regression coefficients for the fixed-effects variables included in the final multivariable model of the log of the daily effective reproduction number (log(Rt)).**

| Variable | Transformation | Regression coefficient (95% CrI) |
|---|---|---|
| Intercept | N/A | 0.108 (0.090, 0.125) |
| Cumulative incidence (sum of cases per 1000 individuals, from the start of an epidemic period up to day t) | Per 1000 people | −0.042 (−0.044, −0.041) |
| Proportion of school holiday days (weekly sum of days of holiday divided by 7, centred around day t) | Proportion | −0.020 (−0.021, −0.018) |
| Maximum temperature (weekly average of daily values centred around day t) | ºC units, divided by 10 | −0.049 (−0.053, −0.045) |
| Cumulative rainfall (weekly sum of daily values centred around day t-14) | mm, divided by 10 | −0.0012 (−0.0014, −0.0010) |
| Maximum relative humidity (weekly average of daily values centred around day t) | Percentage, divided by 10 | 0.010 (0.008, 0.011) |
| EV-A71 proportion (monthly EV-A71 cases/ (EV-A71 + CVA16 cases)) | Proportion | −0.007 (−0.011, −0.003) |

## Factors associated with transmission

To identify factors associated with HFMD transmission we modelled the log of the effective reproduction number (log(Rt)) using linear mixed-effects regression models, with meteorological variables, school holidays, and the proportion of sero-type EV-A71 positive samples as explanatory variables on top of the baseline model. The baseline model accounted for spatiotemporal random effects and the depletion of susceptible individuals using the cumulative incidence of cases during an epidemic as a proxy.

Individual effects (univariable models) of each variable tested and considering different time lagged effects of up to two weeks are shown in S12 Fig. The top three best single predictors of log(Rt) (as determined by the univariable models with the lowest WAIC scores (S2 Table)) were the proportion of school holiday days lagged by zero, one, and two days, which were negatively associated with log(Rt). Among the meteorological variables considered, all temperature variables (maximum, minimum and mean) showed a negative strong association with log(Rt), and an increasing lag backwards in time resulted in an increasingly weaker, but still statistically significant association (S12 Fig).

The final multivariable model (Tables 1 and S3) included the proportion of school holiday days at time *t* (regression coefficient: -0.020, 95% CrI: -0.021, -0.018), and three meteorological variables, which were retained in the following order: maximum temperature at time *t* (-0.049, 95% CrI: -0.053, -0.045), cumulative rainfall at time *t*-14 (-0.0012, 95% CrI: -0.0014, -0.0010), and maximum relative humidity at time *t* (0.0096, 95% CrI: 0.0079, 0.0114). Of the meteorological variables, maximum temperature had the largest effect on log(Rt), and cumulative rainfall and maximum humidity had a very small effect on Rt (S13 Fig). The final model also retained the variable quantifying the proportion of EV-A71 positive samples among all EV-A71 and/or CVA16 positive samples, which was negatively associated with Rt (-0.007, 95% CrI -0.011, -0.003), and it also had a very small effect on Rt (S13 Fig). The depletion of susceptible individuals (-0.042, 95% CrI: -0.044, -0.041), which was included as part of the baseline model, and the proportion of school holiday days had the largest effect on log(Rt) of all the variables included in the final multivariable model (S13 Fig).

The final model reproduced well the general trend of Rt over time, although not the quick and short-term fluctuations (S14–S17 Figs). As a result, the marginal R-squared, which is an estimate of how much variance in Rt is explained by the predictors of the model (i.e., the fixed effects), was low, of only 0.22. However, the model fitted well the magnitude and the trend of Rt, as supported by the analysis of the residuals. Indeed, the residuals were centred around zero (S18 Fig) and the detrended series looked stationary (S19–S22 Figs), with no pattern of trend over time. Spatial and temporal random effects did not show a spatially or temporally correlated structure (S23 Fig).

## Discussion

We analysed the data collected through the national HFMD surveillance system in Malaysia over an 11-year period, from 2009 up to the start of the SARS-CoV-2 pandemic, to describe the spatial and temporal distribution of HFMD cases across the country.

Most years, HFMD incidence was higher in East Malaysia compared to Peninsular Malaysia, and particularly at the start of the timeseries between 2009–2012, which might reflect better surveillance in East Malaysia during the first few years of the study. HFMD incidence also showed a general increasing pattern in Peninsular Malaysia, perhaps reflecting improvements in surveillance over time. Annual incidence rates in the general population (roughly between 500 and 2800 cases per one million people) were similar to those reported from China between 2009 and 2012 (800 – 1600 cases per one million people) [1], which seem to have slightly increased since 2013 based on the comparison of incidence rates per age class [16] (estimates for the whole population were not available in the later study).

In Malaysia, children below six years old represented 85% of all HFMD cases. Among those, the 12–23 months olds were the age class that reported the highest number of HFMD cases (again, in agreement with data from China) [16], with a rapid increase during the first year of life. These observations support current EV-A71 vaccination recommendations in China, of children completing the vaccination schedule before 12 months of age.

Our analyses point to a strong synchronisation of epidemic timing across districts within both Peninsular and East Malaysia, which suggests that there are widespread epidemics in both regions of Malaysia. Despite the observed spatial synchrony, the epidemic peak timing was not regular across years, which demonstrates the absence of a regular seasonal pattern of HFMD in Malaysia. These dynamics are similar to those observed in Singapore, where cases are reported all year-round with some fluctuations [17], and in contrast with Japan [9], China [1], and Hong Kong [10], that have more regular epidemic dynamics. The occurrence of widespread epidemics across the two regions of Malaysia needs a more in-depth study at a more local level that explores the existence of potential spatial waves of spread, for example from urban to rural areas.

The analysis of the temporal drivers of transmission was unable to identify a main factor associated with transmission, other than the depletion of susceptible individuals (which was estimated to reduce Rt by 4.1% for each increase in cumulative incidence of 1 case per 1000 people during an epidemic period). Several meteorological factors (maximum temperature, maximum relative humidity, and cumulative rainfall lagged by 14 days) along with school holidays and the proportion of EV-A71 positive cases were retained in the final multivariable model and were statistically significantly associated with the effective reproduction number, but the magnitude of their respective effects was small. School holidays had the largest effect, with Rt estimated to be reduced by 2% during the school holidays compared to during the school term. It was followed by maximum temperature and cumulative rainfall lagged by 14 days. An increase in the maximum temperature of one degree Celsius was estimated to reduce Rt by 0.48%, and an increase of 10mm in weekly cumulative rainfall to reduce Rt by 0.12%. The last two variables included in the final model were maximum relative humidity and proportion of EV-A71 positive cases, but they did not greatly improve the model fit. We estimated that an increase of maximum relative humidity of 10% increased Rt by 1%, and inversely, that Rt was reduced by 0.7% when all samples were positive for EV-A71 compared to when no samples were positive for EV-A71. Malaysia has a tropical climate and some meteorological variables vary in short ranges throughout the year. For example, the maximum temperature range for the time period and states considered in the regression analysis had a maximum of ten degrees Celsius, and therefore, the total effect of those variables on transmission can be expected to be small. The results of our analyses are consistent with previous analyses from Hong Kong, that found a statistically significant, but limited impact of meteorological factors and school holidays on HFMD transmission [10].

Schools and nurseries are the places where most HFMD transmission is thought to happen and therefore it is not surprising that we estimate a reduction in HFMD transmission during school holidays. It should be noted that the calendar for nursery holidays may not fully coincide with school holidays, and we did not have data on closure of schools or nurseries

as a reactive measure to control HFMD epidemics, which would have allowed us to estimate a possibly stronger effect of closures on HFMD transmission.

Previous studies have shown the importance of the building up of serotype-specific population immunity and subsequent replenishment of susceptible individuals in driving the cyclical patterns of HFMD epidemics [2,9]. However, information on circulating serotypes over the study period was very scarce, not available for all years, and not based on systematic testing. Therefore, we were unable to account for changes in serotype-specific population immunity in the analysis of drivers of transmission. To counter this and as done in previous work [10], we added a variable that measured the cumulative incidence of cases during each epidemic period and included this variable in the baseline model. However, although this approach may account for the depletion of susceptible during a given epidemic, it is unable to capture effects on a longer time scale of years.

We purposely excluded the COVID-19 years in the current manuscript, to avoid confounding effects of the impact of non-pharmaceutical interventions on HFMD transmission. In 2022, Malaysia experienced one of the largest HFMD outbreaks recorded in the country. Future work should focus on quantifying the impact of control interventions against COVID-19 on HFMD dynamics.

To summarise, our analyses suggest that HFMD is endemic in Malaysia, with widespread epidemics, highly synchronised across districts, occurring both within Peninsular and East Malaysia. Factors determining the variability in the timing of HFMD epidemics in Malaysia, and also of regular seasonal patterns in other temperate countries in Asia-Pacific, remain unclear. More work using data at a wider spatial scale covering a diversity of climate regions is needed to identify those.

## Materials and methods

### Ethics statement

This study was approved by Medical Research and Ethics Committee, Ministry of Health, Malaysia (NMRR-ID-22–00201-A4Q, NMRR-19-2149-49617, NMRR-15-1801-28143). Our institution does not require informed consent for retrospective studies of anonymised data.

### Data

**Study area.** Malaysia is comprised of Peninsular Malaysia (bordering Singapore and Thailand) and East Malaysia, which is part of the island of Borneo (with borders to Brunei and Indonesia). There are 13 states and three federal territories (Kuala Lumpur, Putrajaya, and Labuan, herein referred to as states) across Malaysia, which are collectively comprised of 143 districts. There are 13 states in Peninsular Malaysia and three in East Malaysia. Throughout the manuscript, Kuala Lumpur refers to the Federal Territory of Kuala Lumpur.

**HFMD case incidence data.** In Malaysia in 2006, HFMD was made a mandatory notifiable disease under the Prevention and Control of Infectious Diseases Act 1988 [Act 342], which means districts are legally required to report their HFMD cases to the National HFMD Surveillance System through e-Notification [4]. The case definition for HFMD was standardised the same year. Here, we used individual case data recorded by the HFMD case surveillance system of Malaysia between January 1, 2009, and December 31, 2019. Cases are reported when caregivers present children with symptoms compatible with syndromic HFMD, and therefore the data does not include asymptomatic infections. Information used for the analyses included date of symptom onset, age at the time of symptom onset, and district and state of residence. The HFMD case data were kindly provided by the Ministry of Health, Malaysia.

**Population data.** Annual population data for each district was extracted from WorldPop raster files from 2009 to 2019 [18] using administrative level borders defined in GADM shapefiles (https://gadm.org/download_country.html and https://gadm.org/license.html) [19] and the *exactextractr* R package [20].

**HFMD serotype data.** EV-A71 and CV-A16 were typed using a triplex qPCR [21]. The laboratory-confirmed monthly number of EV-A71 and CVA16 positive cases for 2012, 2014, 2015, 2016, 2017, and 2018 (n = 72 months with data) was provided by the National Public Health Laboratory, the Ministry of Health. The monthly proportion of EVA-71 positive cases among all EV-A71 or CVA16 positive cases was calculated, and a spline was fitted through the data points (S1 Fig).

**School holidays.** The dates of school term holidays each year for each state were sourced online from the Ministry of Education, Malaysia (see S2 Fig for the Federal Territory of Kuala Lumpur and Sarawak).

**Meteorological data.** Global two metres above the land surface (2m) dewpoint temperature, 2m temperature, and total precipitation (rainfall) data were downloaded for each hour of each day between January 1, 2012, and December 31, 2019, from the ERA5-Land reanalysis dataset provided by the Copernicus Climate Change Service (ERA5-Land hourly data from 1950 to present) [22], as grid points at 0.1 x 0.1 degrees resolution. Relative humidity ($rh$) was calculated using the dewpoint temperature ($d$) and temperature ($T$) following Equation 1.

$$rh = 100 * \frac{exp(d*17.625)/(d+243.04)}{exp(T*17.625)/(T+243.04)}$$

(1)

A shapefile of Malaysian borders at administrative level one [19] was rasterised to the same resolution as the meteorological data, and used to extract the hourly data for each state in Malaysia. Where borders crossed within a grid point, we assumed the administration unit that uses the largest area of the grid. We calculated the daily cumulative rainfall (mm), the daily average minimum, maximum and mean temperature (ºC) and the daily average minimum, maximum and mean relative humidity, and then averaged over the grid cells in each administration unit (S1 Table and S3 Fig).

## Statistical analyses

**Wavelet analysis of HFMD case timeseries.** We used the *WaveletComp* R package to generate wavelet power spectra of the case timeseries and quantify the periodicity in Peninsular and East Malaysia [23]. We applied a Morlet wavelet to the log(cases+1) transformation of the case timeseries.

**Mean timing of HFMD cases.** We estimated the mean timing of HFMD cases within the year in each state using the total number of cases reported over the entire period of the study. The mean timing of cases within the year was computed using circular statistics with the *circular* R package [24].

**Estimation the effective reproduction number.** We estimated the effective reproduction number, Rt, in each state using EpiFilter [25], which is a method adapted to cases when incidence is low and provides smoother estimates than other classical methods. To remove a potential weekly structure in the data and to obtain smoother estimates of Rt, we applied EpiFilter to the rounded value of the 7-day rolling average of daily incidence. We used a mean serial interval of 3.7 days and a standard deviation of 2.6 days, following Chang *et al.* [10,26], and we set the η parameter of EpiFilter to its default value of 0.1 [25].

**Assessing meteorological and other drivers of transmission.** We estimated the association between transmission (quantified with the effective reproduction number) and a series of meteorological and other factors using estimates from January 1, 2012, to December 31, 2019, as this was the period with observed regular, large-scale outbreaks of HFMD in both East and Peninsular Malaysia. A few states (the Federal Territory of Labuan, Pahang, Perlis, the Federal Territory of Putrajaya, and Terengganu) had long periods of consecutive days reporting zero cases, which led to incomplete continuous timeseries of Rt estimates. We therefore removed these states from the regression analysis. To further remove inconsistent estimates of Rt between epidemic waves (due to low incidence of cases), we performed the regression analysis on the Rt estimates during epidemic periods only, which were determined as follows. We computed the epidemic growth rate on a weekly basis from weekly incidence data, and defined an epidemic period from the week when the growth rate was zero and increasing, up to the first week when the growth rate reached a minimum.

We developed Bayesian mixed-effects linear regression models where the response variable was the median effective reproduction number on the log scale in state $i$ at time $t$, $log(R_{i,t})$, and was assumed to follow a Gaussian distribution. The linear predictor in our baseline model was expressed as shown in Equation 2

$$\log(R_{i,t}) = \alpha + \beta_0 C_{i,t} + \delta_i + \mu_{y,e} \tag{2}$$

where $\alpha$ represents the intercept; $C_{i,t}$ is a proxy for the depletion of susceptibles, which was calculated as the cumulative incidence per 1000 individuals per state for each epidemic period, with regression coefficient $\beta_0$; $\delta_i$ is a spatial random effect on the state $i$, which is modelled as an independent and identically distributed Gaussian random intercept; and $\mu_{y,e}$ is a temporal random effect on the year $y$ separately for Peninsular Malaysia and East Malaysia (indexed by $e$) modelled using a random walk of order one. The random effects aim to capture unstructured noise, for example heterogeneities in reporting rates per state and year.

We tested the association of log(Rt) with the following variables: mean, minimum, and maximum temperature; mean, minimum, and maximum relative humidity; cumulative rainfall; proportion of the week that is school holidays; and proportion of HFMD cases positive for EV-A71 relative to the total number of HFMD cases positive for EV-A71 and/or CVA16 (see S1 Table). We first performed a univariable analysis, where each variable was added to the baseline model one at a time, considering time lags of up to two weeks, to generate 121 univariable models (S2 Table). To calculate the proportion of the week that is school holidays, for each day, $t$, in the timeseries we assigned a binary description if t fell within a school holiday period, and we calculated the proportion of days of holidays in the seven days centred around that day ($t-3$ to $t+3$). Weekends were not treated as school holidays unless they fell strictly within a school holiday period. Similarly, for the meteorological variables, for each day, $t$, we calculated the conditions over $t-3$ to $t+3$, and we re-scaled some of the variables (relative humidity, given in percentage units, was divided by 10; temperature, given in degrees Celsius, was divided by 10; and rainfall, given in millimetres, was divided by 10). We lagged the proportion of school holiday days and meteorological variables each day between $t-1$ and $t-14$.

The best-fitting univariable model, as determined by the lowest Watanabe-Akaike Information Criterion (WAIC) and Deviance Information Criterion (DIC) values (which are estimates of prediction error [27]), was used to develop a multivariable model. We employed a stepwise forward-selection approach, where at each model-building step we selected an additional variable to add to the model, up to the point where the addition of more variables did not improve the WAIC and DIC score by at least four points (S3 Table). Variables were tried in each step if they were not collinear with variables already chosen in a previous step (collinearity was determined as two variables with a Pearson's correlation coefficient $>|0.6|$).

To validate the final model and assess the model fit, we computed the marginal R-squared, which is the proportion of the variance explained by the fixed effects alone relative to the overall variance, and also performed an analysis of the residuals.

We performed a sensitivity analysis testing the meteorological variables as non-linear effects in the univariable regression models. Each variable was categorised according to its 10th to 90th percentiles and added to the baseline model as a random effect categorical variable. The linearity of the regression coefficient pattern over the percentiles was visually assessed (S24 Fig). Linear relationships with HFMD Rt were deemed suitable for the meteorological variables in the main analysis. The regression modelling was conducted in R using the *R-INLA* package [28,29].

## Supporting information

**S1 Fig. Proportion of EV-A71.** The monthly proportion of EVA-71 positive samples among all EVA-71 or CVA16 positive samples (black points) with 95% exact binomial confidence intervals. A spline (shown in red) was fitted through the points using 28 knots (total data points/3).
(PDF)

**S2 Fig. School holidays.** Daily number of hand-foot-and-mouth disease cases reported in the Federal Territory of Kuala Lumpur (top) and Sarawak (bottom) coloured by the school holiday periods in red.
(PDF)

**S3 Fig. Meteorological variables.** Meteorological variables for each day in 2012–2019 coloured for each state in Malaysia, arranged with states in Peninsular Malaysia in the left-hand panels and states in East Malaysia in the right-hand panels.
(PDF)

**S4 Fig. Relationship between district population density and maximum number of consecutive months reporting zero cases.** Districts are coloured according to their state. Population density is taken for 2019. The maximum number of consecutive months with zero reported cases is calculated over the period 2012–2019 for each district.
(PDF)

**S5 Fig. Weekly number of cases.** Weekly number of cases in a log10 scale for all years between 2009 and 2019 in each state.
(PDF)

**S6 Fig. Distribution of cases within the year.** Weekly distribution of annual cases within the year for all years between 2009 and 2019 in each state.
(PDF)

**S7 Fig. Incidence and effective reproduction number of HFMD per state.** (Top) Daily incidence of HFMD cases, coloured by epidemic period (dark green) or not (light green) for Johor, Kedah and Kelantan. (Bottom) Estimated median effective reproduction number with 50% and 95% credible intervals, and epidemic periods shown in dark green.
(PDF)

**S8 Fig. Incidence and effective reproduction number of HFMD per state.** (Top) Daily incidence of HFMD cases, coloured by epidemic period (dark green) or not (light green), for the Federal Territory of Kuala Lumpur, Melaka and Negeri Sembilan. (Bottom) Estimated median effective reproduction number with 50% and 95% credible intervals, and epidemic periods shown in dark green.
(PDF)

**S9 Fig. Incidence and effective reproduction number of HFMD per state.** (Top) Daily incidence of HFMD cases, coloured by epidemic period (dark green) or not (light green), for Perak, Pulau Pinang and Selangor. (Bottom) Estimated median effective reproduction number with 50% and 95% credible intervals, and epidemic periods shown in dark green.
(PDF)

**S10 Fig. Incidence and effective reproduction number of HFMD per state.** (Top) Daily incidence of HFMD cases, coloured by epidemic period (dark green) or not (light green), for Sabah and Sarawak. (Bottom) Estimated median effective reproduction number with 50% and 95% credible intervals, and epidemic periods shown in dark green.
(PDF)

**S11 Fig. Distribution of daily median estimates of the effective reproduction number during epidemic periods in each state considered in the regression analysis.** The dashed black vertical line indicates $R_t = 1$.
(PDF)

**S12 Fig. Estimated regression coefficients (beta value) for each fixed effect variable in the univariable models.** Meteorological and school holiday variables at different lags (day 0–14, on the y-axis), and the proportion of EV-A71

positive samples relative to EV-A71 and/or CVA16 positive samples. The direction of the association is shown in blue for positive (beta>0) and in red for negative (beta<0) associations, and zero (no effect) is shown as a dotted vertical line.
(PDF)

**S13 Fig. Relative effect of each individual variable included in the final model.** Each variable included in the final model was removed, and the Watanabe-Akaike Information Criterion score (WAIC) and the Deviance Information Criterion score (DIC) of the resulting model is shown in red and blue respectively, with the best fitting and final model at the top, with the lowest WAIC and DIC.
(PDF)

**S14 Fig. Model fit to data.** Median estimates of the effective reproduction number during the epidemic periods obtained with EpiFilter are shown in grey, for Johor, Kedah and Kelantan. The estimates obtained with the final mixed-effects regression model are shown in red (median) and pink (95% CrI). We sampled 1000 times from the final model and extracted the 0.50, 0.025, and 0.975 quantiles to estimate the median Rt and the associated 95% CrIs.
(PDF)

**S15 Fig. Model fit to data.** Median estimates of the effective reproduction number during the epidemic periods obtained with EpiFilter are shown in grey, for Kuala Lumpur, Melaka and Negeri Sembilan. The estimates obtained with the final mixed-effects regression model are shown in red (median) and pink (95% CrI). We sampled 1000 times from the final model and extracted the 0.50, 0.025, and 0.975 quantiles to estimate the median Rt and the associated 95% CrIs.
(PDF)

**S16 Fig. Model fit to data.** Median estimates of the effective reproduction number during the epidemic periods obtained with EpiFilter are shown in grey, for Perak, Pulau Pinang and Selangor. The estimates obtained with the final mixed-effects regression model are shown in red (median) and pink (95% CrI). We sampled 1000 times from the final model and extracted the 0.50, 0.025, and 0.975 quantiles to estimate the median Rt and the associated 95% CrIs.
(PDF)

**S17 Fig. Model fit to data.** Median estimates of the effective reproduction number during the epidemic periods obtained with EpiFilter are shown in grey, for Sabah and Sarawak. The estimates obtained with the final mixed-effects regression model are shown in red (median) and pink (95% CrI). We sampled 1000 times from the final model and extracted the 0.50, 0.025, and 0.975 quantiles to estimate the median Rt and the associated 95% CrIs.
(PDF)

**S18 Fig. Histogram of the residuals.** Histogram of the residuals (i.e., difference between observed and mean predicted values) of the final model of log(Rt) for each state.
(PDF)

**S19 Fig. Timeseries of the residuals.** Timeseries of the residuals (i.e., difference between observed and mean predicted values) of the final model of log(Rt) for Johor, Kedah and Kelantan.
(PDF)

**S20 Fig. Timeseries of the residuals.** Timeseries of the residuals (i.e., difference between observed and mean predicted values) of the final model of log(Rt) for Kuala Lumpur, Melaka and Negeri Sembilan.
(PDF)

**S21 Fig. Timeseries of the residuals.** Timeseries of the residuals (i.e., difference between observed and mean predicted values) of the final model of log(Rt) for Perak, Pulau Pinang and Selangor.
(PDF)

**S22 Fig. Timeseries of the residuals.** Timeseries of the residuals (i.e., difference between observed and mean predicted values) of the final model of log(Rt) for Sabah and Sarawak.
(PDF)

**S23 Fig. Spatial and temporal random effects in the final model.** (A) The random effect on year was modelled as a random walk of order 1 for each region, Peninsular and East Malaysia. (B) The random effect per state was modelled using an independent and identically distributed model. The base layer of the map was sourced from GADM (https://gadm.org/download_country.html and https://gadm.org/license.html).
(PDF)

**S24 Fig. Sensitivity analysis testing non-linear relationships between meteorological variables and log(Rt).** Regression coefficients for each percentile of the meteorological variables included as random effect categorical variables in univariable models. The meteorological variables were calculated at time t (lag = 0 days).
(PDF)

**S1 Table. Summary of the variables tested in the mixed-effects regression models.** We lagged the calculated proportion of school holiday days and meteorological variables each day between t and t-14 (n = 15 lags tested in the models).
(DOCX)

**S2 Table. Univariable model variables arranged from top to bottom by the best WAIC score.**
(DOCX)

**S3 Table. Variable selection for final multivariable model** . The final model is shown in bold.
(DOCX)

## Acknowledgments

We acknowledge the Ministry of Health, Malaysia for the provision of data in this work. We acknowledge the use of meteorological data from Muñoz Sabater, J., (2019) [22], which was downloaded from the Copernicus Climate Change Service (C3S) Climate Data Store. The results contain modified Copernicus Climate Change Service information 2020. Neither the European Commission nor ECMWF is responsible for any use that may be made of the Copernicus information or data it contains.

## Author contributions

**Conceptualization:** Yoke Fun Chan, Margarita Pons-Salort.

**Data curation:** Rohani Jahis, Rehan Shuhada Abu Bakar, Noriah Mohd Yusof, Kiroshika Pillay Veel Pilay, Yu Kie Chem, S Selvanesan, Norita Shamsudin, Wes Hinsley, Nor Zahrin Hasran, Norhayati Rusli.

**Formal analysis:** Victoria M Cox, Margarita Pons-Salort.

**Funding acquisition:** Margarita Pons-Salort.

**Investigation:** Margarita Pons-Salort.

**Methodology:** Isobel M Blake, Margarita Pons-Salort.

**Supervision:** Margarita Pons-Salort.

**Validation:** I-Ching Sam, Yoke Fun Chan.

**Visualization:** Victoria M Cox.

**Writing – original draft:** Victoria M Cox, Margarita Pons-Salort.

**Writing – review & editing:** Rohani Jahis, Rehan Shuhada Abu Bakar, Noriah Mohd Yusof, Kiroshika Pillay Veel Pilay, Yu Kie Chem, S Selvanesan, Norita Shamsudin, Wes Hinsley, Isobel M Blake, Nor Zahrin Hasran, Norhayati Rusli, I-Ching Sam, Yoke Fun Chan.

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
