## [Decision Letter · Decision Letter 0]

Response to Reviewers
Revised Manuscript with Track Changes
Manuscript

Shaden Kamhawi

co-Editor-in-Chief

Paul Brindley

co-Editor-in-Chief

**Additional Editor Comments:**

 ?>**Journal Requirements:**

1) Please provide an Author Summary. This should appear in your manuscript between the Abstract (if applicable) and the Introduction, and should be 150-200 words long. The aim should be to make your findings accessible to a wide audience that includes both scientists and non-scientists. Sample summaries can be found on our website under Submission Guidelines:

Potential Copyright Issues:

- Figures 4C, 5B, and S17-B; Please provide a direct link to the base layer of the map (i.e., the country or region border shape) and ensure this is also included in the figure legend; and provide a link to the terms of use / license information for the base layer image or shapefile. We cannot publish proprietary or copyrighted maps (e.g. Google Maps, Mapquest) and the terms of use for your map base layer must be compatible with our CC BY 4.0 license.

5) In the online submission form, you indicated that "will be made available on request following institutional policy governing data sharing". All PLOS journals now require all data underlying the findings described in their manuscript to be freely available to other researchers, either

- In a public repository

- Within the manuscript itself

- Uploaded as supplementary information.

6) Please ensure that the funders and grant numbers match between the Financial Disclosure field and the Funding Information tab in your submission form. Note that the funders must be provided in the same order in both places as well. State what role the funders took in the study. If the funders had no role in your study, please state: "The funders had no role in study design, data collection and analysis, decision to publish, or preparation of the manuscript.".

**Reviewers' comments:**

**Key Review Criteria Required for Acceptance?**

**Methods** :

-Are the objectives of the study clearly articulated with a clear testable hypothesis stated?

-Is the study design appropriate to address the stated objectives?

-Is the population clearly described and appropriate for the hypothesis being tested?

-Is the sample size sufficient to ensure adequate power to address the hypothesis being tested?

-Were correct statistical analysis used to support conclusions?

-Are there concerns about ethical or regulatory requirements being met?

Reviewer #1: (No Response)

Reviewer #2: (No Response)

**Results** :

-Does the analysis presented match the analysis plan?

-Are the results clearly and completely presented?

-Are the figures (Tables, Images) of sufficient quality for clarity?

Reviewer #1: (No Response)

Reviewer #2: (No Response)

**Conclusions** :

-Are the conclusions supported by the data presented?

-Are the limitations of analysis clearly described?

-Do the authors discuss how these data can be helpful to advance our understanding of the topic under study?

-Is public health relevance addressed?

Reviewer #1: (No Response)

Reviewer #2: (No Response)

**Editorial and Data Presentation Modifications?**

Reviewer #1: (No Response)

Reviewer #2: (No Response)

**Summary and General Comments** :

Reviewer #1: This is an interesting epidemiological study that analysed 11 years of surveillance data in Malaysia to investigate the transmission factors of HFMDs, and is well suited to the scope of PLOS Neglected Tropical Diseases. Firstly, I would like to congratulate the authors for piecing together and evaluating an extensive data set. Overall, I very much enjoyed reading this manuscript. The manuscript is well written, proper statistical methods of analysis have been carried out and well explained. I do have a few comments that I hope will help improving the manuscript, or maybe just give the authors food for thoughts.

L174-207: The authors' multivariate analysis is commendable for employing an indicator that precisely quantifies the spread of infectious diseases through the evaluation of Rt. Nevertheless, there remains a concern that the results might vary depending on the methodology used to input the explanatory variables. Ensuring the robustness of the primary findings through comprehensive sensitivity analyses would enhance their reliability. For instance, meteorological factors such as mean temperature, relative humidity, and precipitation could exhibit lag delayed effects. Additionally, it is important to question the assumption of linearity in the exposure-response relationship, thereby justifying the use of a linear mixed-effects regression model.

L174-207: The evaluation of the predictive capability of statistical models is essential. Empirical validation requires an assessment of the model's fit to the data and the control of time-series trends.

L363: While the authors have assumed a Gaussian distribution, it would be prudent to assess the fit against alternative distributions, such as quasi-Poisson or negative binomial, to ensure the most appropriate model is employed.

L385: The examination of delayed effects of meteorological factors should consider a broader range of lags. Given the incubation period of HFMD, an evaluation extending up to 7 to 14 days would be advisable.

L391: If R-INLA is used, it should be evaluated using multiple indices, not just WAIC. For example, it may be necessary to thoroughly examine models with increased complexity regarding input variables and model structure, such as the deviance information criterion and average cross-validation log score.

Dataset and code sharing: It may be a good idea to include the dataset used in this study and the code to reproduce the results in the supplementary materials, so that third parties can reproduce the study. Also, although we were unable to access the Github repository provided by the authors (and so were unable to evaluate its contents), is there a reason for this?

Discussion: It is suggested that a section detailing the study's limitations be included, highlighting three to five key limitations and providing a concise summary of potential avenues for future research.

Reviewer #2: This paper delves into the epidemiological traits of HFMD in Malaysia and scrutinizes the influence of meteorological factors on its prevalence. As a comprehensive study, it holds considerable significance. Nevertheless, there are a few areas for improvement:

1. The dataset spans from 2009 to 2019. Incorporating data from 2020 to 2024 would enrich the analysis, particularly by illuminating the impact of the COVID-19 pandemic on HFMD trends.

2. Conducting a genetic evolutionary analysis of prevalent enteroviruses in Malaysia, such as EV-A71, could provide deeper insights into the virus's mutations and potential implications for HFMD.

PLOS authors have the option to publish the peer review history of their article (what does this mean? ). If published, this will include your full peer review and any attached files.

**Do you want your identity to be public for this peer review?** For information about this choice, including consent withdrawal, please see our Privacy Policy .

Reviewer #1: **Yes: ** Keita Wagatsuma

Reviewer #2: **Yes: ** Yuefei Jin

**Figure resubmission:****Reproducibility:** To enhance the reproducibility of your results, we recommend that authors of applicable studies deposit laboratory protocols in protocols.io, where a protocol can be assigned its own identifier (DOI) such that it can be cited independently in the future. Additionally, PLOS ONE offers an option to publish peer-reviewed clinical study protocols. Read more information on sharing protocols at https://plos.org/protocols?utm_medium=editorial-email&utm_source=authorletters&utm_campaign=protocols

---

## [Decision Letter · Decision Letter 1]

Dear Dr Pons-Salort,

We are pleased to inform you that your manuscript 'Spatio-temporal dynamics of hand, foot and mouth disease in Malaysia, 2009 – 2019' has been provisionally accepted for publication in PLOS Neglected Tropical Diseases.

Best regards,

Eric HY Lau, Ph.D.

Academic Editor

David Safronetz

Section Editor

Shaden Kamhawi

co-Editor-in-Chief

Paul Brindley

co-Editor-in-Chief

Thanks for addressing all the editor’s and reviewers' comments. Congratulations on the excellent work!

Reviewer's Responses to Questions

**Key Review Criteria Required for Acceptance?**

**Methods**

-Are the objectives of the study clearly articulated with a clear testable hypothesis stated?

-Is the study design appropriate to address the stated objectives?

-Is the population clearly described and appropriate for the hypothesis being tested?

-Is the sample size sufficient to ensure adequate power to address the hypothesis being tested?

-Were correct statistical analysis used to support conclusions?

-Are there concerns about ethical or regulatory requirements being met?

Reviewer #1: Improved

Reviewer #2: (No Response)

**Results**

-Does the analysis presented match the analysis plan?

-Are the results clearly and completely presented?

-Are the figures (Tables, Images) of sufficient quality for clarity?

Reviewer #1: Improved

Reviewer #2: (No Response)

**Conclusions**

-Are the conclusions supported by the data presented?

-Are the limitations of analysis clearly described?

-Do the authors discuss how these data can be helpful to advance our understanding of the topic under study?

-Is public health relevance addressed?

Reviewer #1: Improved

Reviewer #2: (No Response)

**Editorial and Data Presentation Modifications?**

Reviewer #1: Accept

Reviewer #2: (No Response)

**Summary and General Comments**

Reviewer #1: Since the authors have earnestly addressed my comments and concerns, I would like to accept this manuscript. I look forward to their future research endeavors and commend the valuable contribution of this study in evaluating the potential application of spatiotemporal statistical models to the transmission dynamics of infectious diseases.

Reviewer #2: (No Response)

PLOS authors have the option to publish the peer review history of their article (what does this mean? ). If published, this will include your full peer review and any attached files.

**Do you want your identity to be public for this peer review?** For information about this choice, including consent withdrawal, please see our Privacy Policy .

Reviewer #1: **Yes: ** Keita Wagatsuma

Reviewer #2: No

---

## [Editor Report · Acceptance letter]

Dear Dr Pons-Salort,

We are delighted to inform you that your manuscript, "Spatio-temporal dynamics of hand, foot and mouth disease in Malaysia, 2009 – 2019," has been formally accepted for publication in PLOS Neglected Tropical Diseases.

Best regards,

Shaden Kamhawi

co-Editor-in-Chief

Paul Brindley

co-Editor-in-Chief
